# Applications of Programmable Endonucleases in Sequence- and Ligation-Independent Seamless DNA Assembly

**DOI:** 10.3390/biom13071022

**Published:** 2023-06-21

**Authors:** Xingchen Xiong, Zhiwen Lu, Lixin Ma, Chao Zhai

**Affiliations:** State Key Laboratory of Biocatalysis and Enzyme Engineering, Hubei Collaborative Innovation Center for Green Transformation of Bio-Resources, School of Life Sciences, Hubei University, Wuhan 430062, China; 202031107021057@stu.hubu.edu.cn (X.X.); 202021107011172@stu.hubu.edu.cn (Z.L.); malixing@hubu.edu.cn (L.M.)

**Keywords:** DNA assembly, programmable endonuclease, homologous recombination

## Abstract

Programmable endonucleases, such as Cas (Clustered Regularly-Interspaced Short Repeats-associated proteins) and prokaryotic Argonaute (pAgo), depend on base pairing of the target DNA with the guide RNA or DNA to cleave DNA strands. Therefore, they are capable of recognizing and cleaving DNA sequences at virtually any arbitrary site. The present review focuses on the commonly used in vivo and in vitro recombination-based gene cloning methods and the application of programmable endonucleases in these sequence- and ligation-independent DNA assembly methods. The advantages and shortcomings of the programmable endonucleases utilized as tools for gene cloning are also discussed in this review.

## 1. Introduction

DNA assembly is the primary step in basic research, gene engineering, and synthetic biology [1]. Multiple DNA modules need to be cleaved precisely and assembled simultaneously in most cases to construct new metabolic pathways in the host cells. The conventional methods of restriction enzyme cutting and ligase re-ligation depend on specific sequences in DNA fragments. The whole procedure is time-consuming and expensive. Moreover, it is very difficult to find unique recognition sites for restriction enzymes when facing large DNA fragments or handling high-throughput DNA assembly. Aiming for large-scale DNA cloning at a low cost, many sequence-independent DNA cloning methods were developed, such as Gateway cloning [2,3], mating-assisted genetically integrated cloning (MAGIC) [4], Golden gate cloning [5,6,7], etc. Among them, sequence- and ligation-independent DNA cloning methods based on in vitro and in vivo homologous recombination, including SLIC, In-fusion, ET cloning, etc., became popular. In recent years, programmable endonucleases have been employed to cleave DNA precisely to specific bases and facilitate the preparation of DNA fragments. Nowadays, gene assembly is incredibly precise, fast, and efficient because of these remarkable DNA assembly methods and tool enzymes. In the present review, we introduce the development of DNA cloning methods based on in vitro and in vivo homologous recombination and the application of programmable endonucleases in DNA assembly.

## 2. In Vitro Cloning Based on Homologous Recombination

In 1990, a ligation-independent cloning (LIC) method was reported [8]. This LIC-PCR method introduced homologous recombination into in vitro gene cloning for the first time. Both the insert and vector are amplified with PCR. The primers for PCR are designed carefully. The 5′-ends of the primers used to amplify the insert contain an additional 12-nt sequence lacking dCMP. As a result, the amplification products include 12-nt sequences lacking dGMP at their 3′ ends. The 3′-terminal sequences can be removed by the 3′-5′ exonuclease activity of T4 DNA polymerase in the presence of dGTP, leading to 5′-overhangs of a defined sequence and length. On the other hand, the vector is inversely amplified with primers containing a region compatible with the sequence in the multiple cloning site and an additional 12-nt tail complementary to the sequence at the ends of the insert, permitting the creation of single-stranded ends with T4 DNA polymerase in the presence of dCTP. The insert and vector backbone are co-transformed into *E. coli* after digestion and circularization are mediated by the 12-nt overhangs at the ends of the vector backbone and the insert. The resulting circular recombinant molecules do not require in vitro ligation for efficient bacterial transformation. The nicks are repaired by *E. coli*. However, this method is rather inconvenient because both the vector and insert are amplified with long primers. Li et al. simplified this method and developed sequence and ligation-independent cloning (SLIC) in 2007 [9,10]. With SLIC, the cloning vector is linearized with restriction enzymes or inverse PCR, while the insert is amplified with PCR to introduce two short homologous ends complementary to the linearized vector. The insert and linearized vector are mixed and treated with T4 polymerase, which exerts its 3′-5′ exonuclease activity to form single-stranded regions at the homologous ends. The digestion is terminated by adding a single dNTP, followed by transformation into *E. coli*. Although gaps are left in the recombination intermediate due to the polymorphism of the single-stranded regions, the gaps can be fixed efficiently in vivo to gain recombinant plasmids. Up to 10 DNA fragments were assembled with this method. Moreover, this method established a new avenue for gene cloning. In the same year, the In-fusion method was published [11]. The principle of In-fusion is similar to that of SLIC. However, instead of T4 polymerase, the In-Fusion Dry-Down reaction mix contains poxvirus DNA polymerase, which also has 3′-5′ exonuclease activity in the presence of Mg^2+^ and low concentrations of dNTP [12]. To date, In-fusion is still a popular gene cloning method. It is widely used in the construction of CRISPR/Cas (Clustered Regularly Interspaced Short Repeats and CRISPR-associated proteins) expression vectors [13] and plasmid libraries [14], etc. In 2012, Seamless Ligation Cloning Extract (SLiCE) was developed [15,16]. SLiCE does not require enzymes for the modification of vector and insert end sequences (e.g., T4 DNA polymerase) or ligases. This method utilizes in vitro homologous recombination activities in cell lysates prepared from RecA^−^ *E. coli* to assemble multiple DNA fragments bearing homologous ends >15 nt in a single recombination reaction. To improve the cloning efficiencies, *E. coli* DH10B expressing an optimized λ prophage Red recombination system was used as the host. Subsequently, Motohashi demonstrates that the endogenous RecA-independent recombination activities in *E. coli* RecA^−^ laboratory strains can function efficiently for SLiCE using short homology lengths (approximately 15–19 bp) without the requirement of exogenously expressing λ prophage Red/ET recombination system. Multiple fragments were cloned successfully using SLiCE with cell lysate carefully prepared from the late log phase cells of *E. coli* RecA^−^ laboratory strains [17,18].

With in vitro cloning, DNA polymerases with 3′–5′ proofreading activity are chosen to generate single-stranded homologous ends for in vitro intermolecular annealing of DNA fragments. However, the exonuclease activities of these polymerases are relatively low. Other enzymes are utilized to improve the efficiency of generating suitable overhangs. One of them is uracil DNA glycosidase (UDG) in the uracil-specific excision reagent (USER) cloning technique. PCR primers containing approximately 12-nt 5′ tails in which at least four deoxyuridines have been placed instead of deoxythymidines are designed to amplify the insert. After PCR, the product is treated with UDG, which excises uracil bases selectively and leaves the phosphodiester backbone intact. The abasic sites destabilize base pairing, which in practice results in 3′ overhangs that can anneal to the vector backbone produced in the same way. Alternatively, the backbone can also be prepared by ligating a short U-containing DNA sequence to the linearized vector, followed by treatment with UDG. The treated PCR product and vector backbone are able to form a stable circular hybridization product used to transform *E. coli* without prior ligation [19]. In 1997, Watson and Bennett upgraded this method by adding a second enzyme, T4 endonuclease V, which is capable of breaking the DNA phosphodiester backbone at the 3′ side of an abasic site [20]. In 2006, Halkier et al. demonstrated that a proof-reading DNA polymerase, PfuCx, is compatible with USER and improved the accuracy of this method [21,22]. Nowadays, the commercial USER™ enzyme (New England Biolabs, Cambridge, UK) consists of UDG and DNA glycosylase-lyase endo VIII. On the other hand, Gibson assembly uses T5 exonuclease to fulfill this goal. Three enzymes, T5 exonuclease, DNA polymerase, and DNA ligase, are employed for this method. T5 exonuclease cleaves linear DNA molecules in a 5′ to 3′ direction to form long overhangs up to hundreds of base pairs for specific annealing between DNA fragments. Phusion DNA polymerase is employed to fill the gaps after the annealing of single-stranded DNA-compatible regions. Next, Taq DNA ligase joins the DNA fragments [23,24]. Gibson assembly is highly efficient in the DNA cloning of large fragments up to thousands of kilobase pairs [25]. However, T5 exonuclease is a double-bladed sword. Its strong activity (approximately 10–30 nt per second at 37 °C [26]) facilitates the formation of long single-stranded regions at the termini of the DNA fragments, which in turn promotes recombination efficiency. On the other hand, it makes Gibson assembly unsuitable for cloning DNA fragments shorter than 200 bp since the whole insert fragment may be degraded completely after T5 exonuclease treatment. Therefore, it is critical to slow down T5 exonuclease for the cloning of small fragments. The optimal temperature for T5 exonuclease is 37 °C [26]. Gibson chose an unfavorable temperature of 50 °C to generate the isothermal process matching Phusion DNA polymerase and Taq ligase. Meanwhile, it also partially jeopardized the high activity of T5 exonuclease to avoid overtreatment. Our group adopted another strategy to decrease the activity of T5 exonuclease. Through ten-fold dilution of the enzyme and setting the reaction temperature at 0 °C, the cleavage speed of T5 exonuclease was slowed down to approximately 3 nt per min [27,28]. Therefore, single strands of approximately 15 nt were generated in 5 min, which is long enough for efficient recombination. On the other hand, this modest size avoids low recombination efficiency caused by the stable secondary structure formed by long sticky ends [29]. In another report, T5 exonuclease DNA assembly (TEAD) was established, and the activity of T5 exonuclease was controlled through dilution to 0.04 U per reaction and the addition of PEG8000 [30]. With these improvements, T5 exonuclease-mediated gene cloning becomes extremely simple, especially for vectors smaller than 3 kb. The insert and vector backbone were amplified with PCR. Two homologous regions of 15–25 bp compatible with the ends of the vector backbones were introduced into the inserts with primers. The PCR products were mixed without purification and treated with 0.5 U of T5 exonuclease at 0 °C for 5 min, followed by transformation into *E. coli* to generate recombinant plasmids.

## 3. In Vivo Cloning Based on Homologous Recombination

Besides the in vitro cloning methods, more controversial bacterial in vivo DNA cloning methods were developed at the end of the last century based on the phenomenon that some strains of *E. coli* can take up linear vectors, insert fragments, and assemble them in vivo when the ends of the linear DNA fragments contain 20 to 50 bp of overlapping homologous sequences [31]. As early as 1993, Bubeck et al. co-transformed a fragment of 1.1 kb bearing homologous regions with a linearized pBluescript SK(-) vector into *E. coli* DH5α and gained transformants harboring recombinant plasmids with the insert assembled in the correct orientation [32]. In 1998, Zhang et al. demonstrated DNA assembly between linear and circular DNA. The linear insert was prepared with PCR using 60-nt oligonucleotides consisting of 42 nt of homology to chosen regions in a vector and 18 nt at the 3′ ends complementary with the target gene. The linear insert and circular vector were co-transformed into a variety of *E. coli* hosts. Recombinants were obtained in *sbcA E. coli* expressing the Rac phage protein pair, RecE and RecT. Therefore, this method was named “ET cloning” [33]. RecE and RecT are 5′–3′ exonucleases and annealing proteins, respectively [34]. Later, the same group demonstrated direct cloning of chosen DNA regions from complex mixtures of genomic DNA using ET cloning. The efficiency decreased as the complexity of these genomes increased [35]. ET cloning can also be carried out in *E. coli* expressing Redα/Redβ, the functional counterparts of RecE/RecT in λ phage. The cloning efficiency is even higher after the replacement [36].

Although bacterial in vivo cloning is more straightforward than in vitro methods, bacterial in vivo cloning is not as popular as in vitro cloning approaches. One reason is that in vivo cloning depends on highly competent cells (in general, 10^6^–10^7^ cfu/μg of DNA for single or double recombination events and 10^7^–10^9^ cfu/μg for three or more recombination events) [37]. Another reason is that the recombination mechanism of in vivo gene cloning is poorly understood [38]. It is generally assumed that bacterial in vivo cloning requires *E. coli* strains with enhanced recombination ability. Recombinases, such as endogenous RecA or phage-derived RecE/RecT, form recombinant DNA by fusing fragments using homologous sequences between different DNA molecules. On the other hand, the ability of these enzymes to recombine DNA also causes plasmid instability, inducing deletions, multimerization, or genomic integration of plasmid sequences. For this reason, “recombination-deficient” bacterial strains lacking RecA or RecE/RecT expression are ubiquitously used for the growth and maintenance of plasmid DNA for routine laboratory work. Therefore, it causes a conflict. The generally used RecA^-^ strains cannot be used as hosts for bacterial in vivo cloning. However, subsequent research indicated that in vivo homologous recombination utilizing the RecA-independent recombination pathway commonly present in laboratory *E. coli* strains was exploited. The result indicated that *E. coli* XL-10 Gold, DH5α, Mach1, and Stbl3 strains are suitable for in vivo cloning [39].

Meanwhile, the mechanism of in vivo cloning was clarified. From their work on plasmid re-circularization, Conley et al. proposed that RecA-independent recombination occurred through a single-stranded annealing mechanism. In this model, in vivo 3′ to 5′ exonuclease activity would produce ssDNA at linear ends, which can anneal through short regions of homology to be subsequently repaired by polymerases and ligases [40]. Consistent with this theory, Nozaki discovered that the in vivo cloning of *E. coli* is independent of both RecA and RecET recombinases but is dependent on XthA, a 3′–5′ exonuclease. Accordingly, they developed in vivo *E. coli* cloning (iVEC). The same report also indicated that iVEC activity is reduced by the deletion of the C-terminal domain of DNA polymerase I (Pol A). Collectively, these results suggest the following mechanism for iVEC: Firstly, XthA resects the 3′ ends of linear DNA fragments that are introduced into *E. coli* cells, resulting in the exposure of the single-stranded 5′ overhangs. Next, the complementary single-stranded DNA ends hybridize each other, and gaps are filled by DNA polymerase I. Multiple-fragment assembly of up to seven fragments was demonstrated in combining iVEC with a modified host strain, sN1187, bearing the Δ*hsdR* Δ*endA*Δ*recA* triple deletion [41].

Based on this theory, the key step for both in vitro and in vivo cloning involves the formation of single-stranded DNA terminals through exonucleases, which are added artificially in vitro or obtained from an endogenous source. Therefore, the procedures of both strategies are unified in the end (Figure 1). Hence, recombinant vectors may be obtained through both in vivo and in vitro models, regardless of particular methods.

## 4. Application of CRISPR/Cas in DNA Assembly

Regardless of in vitro or in vivo DNA assembly, DNA fragments must be linear to facilitate recombination. The commonly used strategies involve restriction enzyme digestion and PCR. However, it is difficult to find unique restriction sites for precise cleavage of the DNA fragments. On the other hand, PCR may introduce mutations into the fragments. Moreover, utilizing inverse PCR to linearize the cloning vector is limited by the difficulty of amplifying vectors with high GC content, repeats, or long sequences. As in the bacterial immune system, programmable endonucleases from the CRISPR/Cas system depend on the base pairing of the target DNA with the guide RNA to cleave the double-stranded genome of viruses [42,43,44,45]. The high specificity and catalytic activities make these enzymes ideal tools for DNA editing [46,47,48] and transcription regulation [13,49,50,51]. Moreover, they are remarkable alternatives to restriction enzymes because the Cas/sgRNA complexes induce highly precise double-stranded breaks at virtually every nucleotide in a DNA fragment.

## 5. Application of CRISPR/Cas9 in DNA Assembly

Cas9 belongs to the class 2 type II CRISPR/Cas system [52]. The Cas9 protein has approximately 800 to 1400 amino acids and forms a two-lobed structure (Figure 2A), with the target DNA and sgRNA positioned in the interface between the two lobes. Two loops in both lobes contribute to the recognition of the PAM. A conserved arginine cluster at the N-terminus of Cas9 belongs to a bridge helix, which is critical for sgRNA. Cas9 contains two nuclease domains: the RuvC nuclease (RNase H fold) domain near the amino terminus and the HNH nuclease domain that is located in the middle of the protein. They are required for target recognition and DNA cleavage [53,54]. The Cas9/sgRNA complex recognizes a 17–20 base target site, which can be of any sequence as long as it is located 5′ of the protospacer adjacent motif (PAM) (Figure 2A). Thus, it can be programmed to cleave almost anywhere with a stringency higher than that of one cleavage in a sequence of human genome size [45].

In 2015, a Cas9/sgRNA complex was applied to linearize a 22-kb vector (pLACAGRFPTetOn), and subsequently, a 783-bp DNA fragment was cloned into this large vector using Gibson assembly [56]. In the same year, two other papers were published about cloning large DNA fragments up to hundreds of kilobase pairs by combining Cas9 cleavage with Gibson assembly or homologous recombination in yeast. Cas9-Assisted Targeting of Chromosome segments (CATCH) utilizes the gel method [57,58]. Both the lysis of bacteria cells and Cas9/sgRNA cleavage of the chromosomes are carried out in agarose gel to generate large fragments up to a hundred kilobase pairs. These gigantic DNA fragments are mixed with cloning vectors bearing 30-bp terminal sequence overlaps with the target DNA at both ends, followed by Gibson assembly. The recombinant plasmid is then electrotransformed into a cloning host. On the other hand, the CRISPR/Cas9-mediated transformation-associated recombination (TAR)-CRISPR protocol is performed in solution [59]. With the conventional TAR method, the genomic DNA is extracted and randomly sheared to gain a population of overlapping DNA fragments. Furthermore, the genomic DNA fragments are co-transformed into yeast cells along with a vector carrying gene-specific targeting sequences at both ends (hooks). Upon co-transformation into yeast, homologous recombination occurs between the vector’s hooks and targeted genomic sequences flanking the gene of interest to form a circular Yeast Artificial Chromosome (YAC). This YAC readily propagates, segregates, and can be selected for in yeast [60]. The efficiency of TAR is only 0.5–2% because the distance between the homologous sites and the DNA ends varies between DNA fragments, which in turn affects the homologous recombination between the inserts and vectors. TAR-CRISPR increased the cloning efficiency up to approximately 32% by introducing more accurate double-stranded breaks at the homologous sites in the desired genomic fragments with Cas9/sgRNA cleavage (Figure 3). In 2018, the ExoCET method was developed, combining Cas9 cleavage of genomic DNA with single-stranded formation using T4 polymerase and ET cloning through overexpression of RecE/T or Redα/Redβ in host strains. This method was successfully applied to clone large regions (>50 kb) from bacterial and mammalian genomes directly, and the size of the insert reached >100 kb [61].

Another strategy for the cloning of large gene clusters couples Cas9 with an in vitro λ packaging system. The genomic DNA of bacteria was sheared randomly, and the fragments were dephosphorylated, followed by digestion with the Cas9/sgRNA complex at specific sites to gain DNA fragments with 5′-phosphate groups. Only DNA fragments bearing 5′-phosphate groups were capable of being ligated with vectors bearing 5′-OH and subsequently packaged into phage particles by an in vitro λ packaging system to infect *E. coli*. With this method, the natural product pathways of Tü3010 (27.4 kb) and sisomicin (40.7 kb) were cloned from the bacterial genome [62].

Meanwhile, to develop more efficient tools for DNA cleavage, Cas9 was optimized through protein engineering. Cas proteins require PAMs to cleave their targets. For instance, the PAM of Cas9 *Streptococcus pyogenes* (SpCas9) is 5′-NGG-3′ [63]. PAMs limit the target site recognition of Cas to a subset of sequences. Therefore, many variants of Cas9 were constructed to eliminate the limitations of PAMs [64]. Among them, Cas9 SpRY is a near-PAMless SpCas9 variant (bearing A61R, L1111R, N1317R, A1322R, and R1333P substitutions). Cas SpRY exhibits robust activity on a wide range of sites with non-canonical NRN (R stands for A or G) PAMs and lower but substantial activity on those with NYN (Y stands for C or T) PAMs in human HEK 293T cells [65]. Moreover, SpRY was discovered to be PAMless in vitro and can cleave DNA at practically any sequence. With the double gRNA SpRYgests strategy, this variant can overcome the weak off-target cleavage and the activity inhibition by specific sequences (for instance, an NYN PAM with a 1st spacer C), hence causing precise double-stranded breaks in DNA fragments during DNA cloning and mutagenesis [66].

## 6. Application of CRISPR/Cas12a in DNA Assembly

Another Cas protein, Cas12a, is also employed for DNA assembly in vitro. In 2013, the class 2 type V CRISPR/Cas system, which contains Cas12a (CRISPR from *Prevotella* and *Francisella*, also known as Cpf1), was identified [67]. Subsequently, Cas12a was characterized in 2015 [68]. The size of Cas12a is approximately 1300 amino acids. Unlike Cas9, Cas12a has only the RuvC catalytic domain to guide RNA-directed dsDNA cleavage [55] (Figure 2B). Cas12a recognizes a short T-rich region (5′-YTN-3′) and cleaves double-stranded DNA, releasing DNA fragments with 5′-overhangs of 4–7 nt (Figure 2B). Apparently, the PAM of Cas12a is completely different from the G/C-rich PAM of Cas9. Therefore, researchers tried to develop in vitro gene cloning methods using Cas12a to overcome the PAM limitation of Cas9. Shi et al. developed the C-brick method based on Cas12a for high-throughput assembly of expression cassettes [69]. However, the recombinant efficiency was only 25–30%, which was much less than the recombinant efficiency of the RE-based method (around 70–80%). A following paper from the same group found out that Cas12a cleaves randomly at the 13th, 14th, 18th, and 19th bases next to PAM on the non-target strand and the 21st to 24th bases on the target strand. The highly polymorphic sticky ends generated by Cas12a are unsuitable for ligation in the same manner as restriction enzymes. They solved this problem by adding Taq ligase, which only recognizes accurately paired sticky ends. In combination with carefully designed crRNAs to improve the accuracy of Cas12a cleavage, the recombination efficiency increased to 70–80% [70]. Our group coupled FnCas12a with T5 exonuclease and developed the CT5 cloning method in 2020 [71]. FnCas12a cleaved donor and recipient vectors and released DNA fragments bearing homologous regions at both ends. Subsequently, T5 exonuclease is employed to generate long single-stranded DNA ends (20–30 nt) for homologous annealing. The DNA fragments are co-transformed into *E. coli,* and the double-stranded breaks are repaired in *E. coli* host cells to gain recombinant vectors. Moreover, this combination was also applied to site-directed mutagenesis, achieving mutations specific to each base pair in a vector [72].

Cas12a was also applied for the cloning of large gene clusters. Cas12a-assisted precise targeted cloning using in vivo *Cre-lox* recombination (CAPTURE) consists of Cas12a digestion, a DNA assembly approach termed T4 polymerase exo+ fill-in DNA assembly, and the Cre-lox in vivo DNA circularization system. Purified genomic DNA is digested by Cas12a to release the target fragments. Next, two DNA receivers containing *lox*P sites at their ends were added to the ends of the target fragments using the exonuclease and fill-in activity of T4 DNA polymerase. In the final step, the assembly mixture is transformed into *E. coli* harboring a circularization helper plasmid. Linear DNA is circularized in vivo by Cre-*lox* recombination. With this method, 47 BGCs (biosynthetic gene clusters) ranging from 10 to 113 kb from both Actinomycetes and Bacilli were cloned with ~100% efficiency [73]. Meanwhile, CRISPR/Cas12a-mediated fast direct biosynthetic gene cluster cloning (CAT-FISHING) was also developed to directly capture large BGCs, combining Cas12a and bacterial artificial chromosome library construction. Two homology arms (each containing at least one PAM site, or ≥30 bp) that flank the target BGC were selected as adapter sequences. A bacterial artificial chromosome containing these two adapter sequences was constructed as the capture plasmid and linearized with Cas12a. Meanwhile, genomic DNA was digested with the Cas12a/sgRNA complex. Next, the linearized capture plasmid and the digested genomic DNA were mixed and ligated with DNA ligase. Furthermore, the ligation product was introduced into *E. coli* by electroporation. The successfully captured BGCs were identified through PCR. Several large BGCs from different actinomycetal genomic DNA samples were efficiently captured by CAT-FISHING, the largest of which was 145 kb with 75% GC content [74].

## 7. Application of Prokaryotic Argonaute Proteins in DNA Assembly

Prokaryotic Argonaute proteins (pAgos) belong to another group of programmable endonucleases [75]. Ago proteins were initially discovered in eukaryotic cells and are the key elements of RNA interference. As the functional core of the RNA-induced silencing complex (RISC), eukaryotic Argonaute proteins (eAgos) use small single-stranded RNA molecules as a guide to target complementary RNA sequences [76]. Ten years later, their counterparts in prokaryotes, pAgos, were discovered in archaea and bacteria [77,78]. pAgo proteins show higher diversity in structure and features in comparison to eAgos. In 2018, 1010 putative pAgos were predicted from 1385 completely and partially sequenced genomes of bacteria and archaea using known pAgos as queries by PSI-BLAST [79]. Moreover, the structures of pAgo also show high diversity. eAgos consists of six domains, including the N-terminal domain, linker1, PAZ (PIWI-Argonaute-Zwille), linker2, MID (middle), and PIWI (P element-induced wimpy testis). All known pAgos contain the PIWI domain. However, many pAgos lack an N-terminal domain and a PAZ domain, while others have a nuclease or DNA-binding domain. Accordingly, pAgos are classified as long pAgos, short pAgos, and PIWI-RE proteins [77,80]. The structure of a pAgo from *T. thermophilus* (*Tt*Ago) is illustrated in Figure 4 to display the typical structure of a long pAgo [81]. Many pAgos are capable of recognizing and cutting DNA targets when loaded with a complementary DNA guide. For instance, *Tt*Ago cleaves DNA under the guidance of a 5′-phosphorylated oligonucleotide of 13 to 25 nt (Figure 4). Moreover, they are independent from PAM motifs, which makes them ideal tools for gene cloning [82].

In 2017, Enghiad and Zhao developed artificial restriction enzymes (AREs) based on *Pyrococcus furiosus* Argonaute (*Pf*Ago) [83]. Short DNA guides bearing 5′-phosphorylated groups direct *Pf*Ago to target sites for cleavage at high temperatures (>87 °C). These AREs are capable of recognizing and cleaving DNA sequences at virtually any arbitrary site and generating defined sticky ends of 2–5 nt, which is similar to restriction enzymes. Eighteen AREs were generated for DNA fingerprinting and molecular cloning of PCR-amplified or genomic DNA. These AREs work as efficiently as their naturally occurring counterparts, and some of them even do not have any naturally occurring counterparts. The same group developed a nearly fully automated workflow named PlasmidMaker that allows error-free construction of plasmids with virtually any sequence in a high-throughput manner. This platform consists of a DNA assembly method using *Pf*Ago or its variant [76]-based artificial restriction enzymes, a user-friendly frontend for plasmid design, and a backend that streamlines the workflow and integrates with a robotic system. With this platform, 101 plasmids from 6 different model organisms ranging in size from 5 to 18 kb were generated [84]. For the time being, there is no report about coupling pAgos cleavage with homology-based DNA assembly methods. One reason is because pAgos generate single-stranded DNA with a predefined length suitable for ligation. On the other hand, the non-specific cleavage of pAgos causes low recombination efficiency during co-transformation without ligation.

## 8. Conclusions

The combination of programmable endonucleases with homologous recombination makes gene cloning as precise as medical operations. It also facilitates the cloning of large BGCs up to Mb (Table 1). However, there is still a long way to go before these methods become routine tools in laboratories. For the CRISPR/Cas system, the key is the commercial availability of sgRNAs. For instance, the sgRNA of Cas9 is approximately 100 nt. Chemistry-based synthesis of long RNAs is too expensive for the time being, while biosynthesis with in vitro transcription using oligonucleotides as templates is inconvenient. Moreover, the cleavage efficiency needs to be improved through protein engineering and gRNA engineering. As mentioned above, the limitation of PAM needs to be eliminated through protein engineering for versatile and precise DNA digestion. Moreover, the choice of sgRNAs also affects the cleavage efficiency of Cas/sgRNA complexes through failure to form functional complexes or inability to recognize targets blocked by complex chromatin structures [85].

On the other hand, the application of pAgos in gene cloning is still in the initial stages. Although pAgos can use short, stable, and cheap DNA guides and the cleavage is independent from PAM, there is a bottleneck for the application of pAgos in gene cloning. pAgos cleaves single-stranded DNA without secondary structure. Therefore, for the time being, only thermophilic pAgos, such as *Pf*Ago, *Tt*Ago, etc., are applied to generate double-stranded breaks in plasmids at high temperatures at which DNA double strands are dissociated. Although pAgos were found in mesophilic bacteria, they are unable to cleave double-stranded DNA at their optimal temperature [86,87] and have not been applied to gene cloning yet. In 2022, Vaiskunaite et al. demonstrated efficient cleavage of linear double-stranded DNA molecules with a combination of *Cb*Ago and RecB^exo–^C. *Cb*Ago is a mesophilic Argonaute derived from *Clostridium butyricum,* while RecB^exo–^C is a mutant of RecBC DNA helicase, remaining unwinding activity but lacking nuclease activity. This combination can cleave specific sites located 11 to 12.5 kb from the ends of linear double-stranded DNA molecules at 37 °C. Since the RecB^exo–^C DNA helicase cannot unwind circular DNA, the *Cb*Ago/RecB^exo–^C system fails to cleave circular DNA. However, this method shows a novel avenue for in vitro DNA assembly using pAgo [88]. Another shortcoming of pAgos is its strong sequence bias [89]. Most DNA sequences cannot be cleaved, and the mechanisms that determine sequence recognition and cleavage efficiency remain unknown. Therefore, pAgos with high activity and specificity need to be generated with protein engineering to meet the requirements of highly efficient gene cloning methods.

In summary, the present review introduces the application of programmable endonucleases in sequence- and ligation-independent DNA assembly. More convenient and efficient methods may be developed with further study of these enzymes.

## Figures and Tables

**Figure 1 biomolecules-13-01022-f001:**
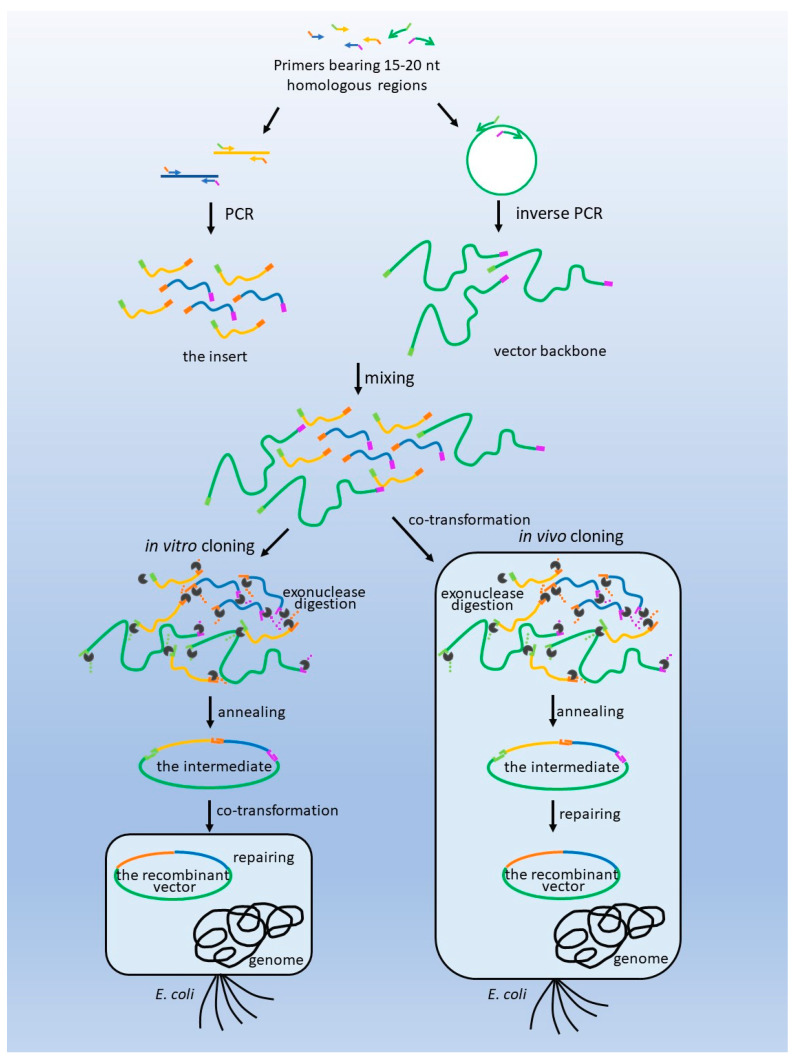
**The strategy of in vitro and in vivo DNA cloning methods.** Both in vitro and in vivo cloning involve the formation of single-stranded DNA terminals through exonucleases, the annealing of complementary ends to form the intermediate, and the repair of the gaps to form recombinant vectors in host cells.

**Figure 2 biomolecules-13-01022-f002:**
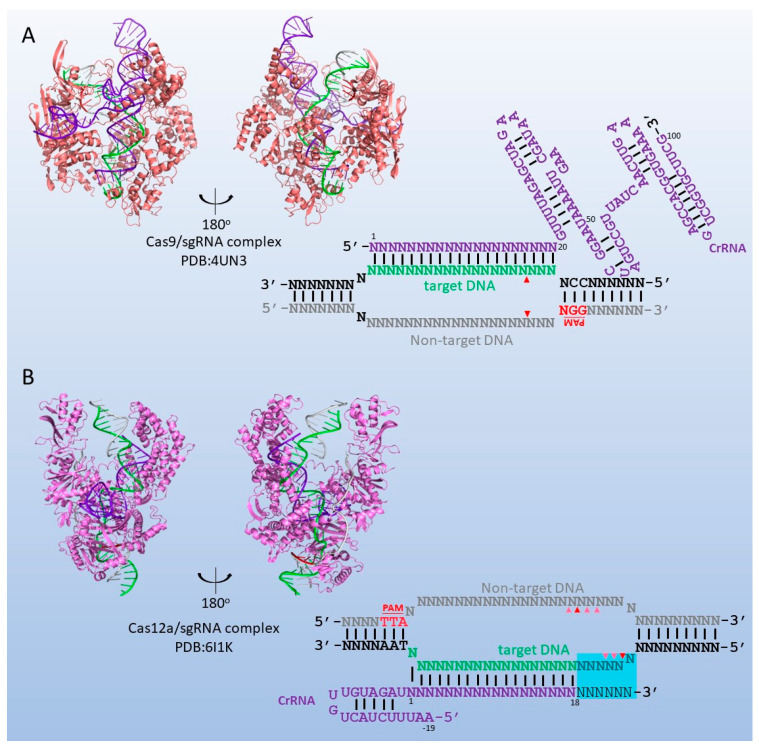
**The structure of Cas9 and Cas12a and the structure of the cleavage sites.** (**A**). The structure of Cas9 (PDB:4UN3) [54] and the cleavage site. (**B**). The structure of Cas12a (PDB:6I1K) [55] and the cleavage site. The non-target DNA strands are labeled gray. The target DNA strands are labeled green. The sgRNAs are labeled purple. PAMs and cleavage sites are labeled in red. The wobble sequence without the requirement of the strict complement is highlighted in blue.

**Figure 3 biomolecules-13-01022-f003:**
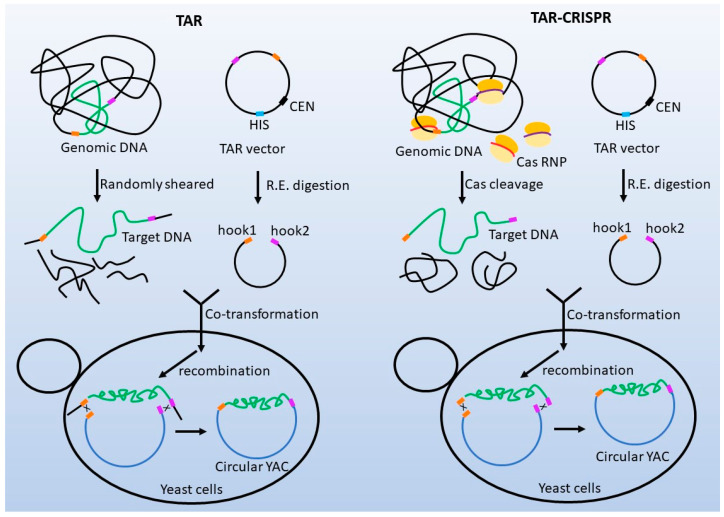
**Cloning of BGCs with or without coupling programmable endonucleases.** With the traditional TAR [60], genomic DNA is randomly sheared and co-transformed into yeast with a vector linearized with restriction enzymes. Therefore, the recombination does not occur exactly at the ends of the insert, which decreases the recombination efficiency dramatically. On the other hand, Cas RNPs cleave the genomic DNA at the desired sites in (TAR)-CRISPR [59], hence the recombination happens at the ends of the insert and vector backbone to guarantee high cloning efficiency.

**Figure 4 biomolecules-13-01022-f004:**
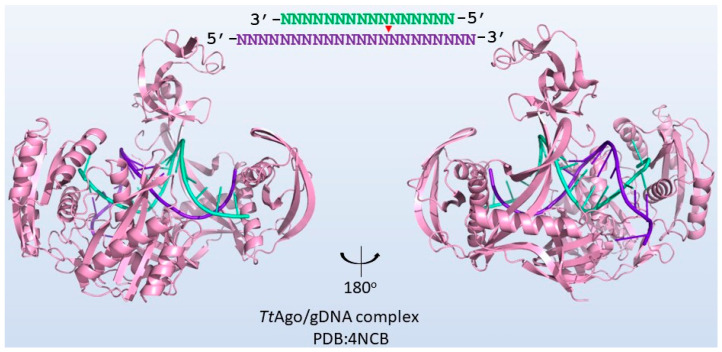
**The structure of *Tt*Ago (PDB:4NCB) [81] and the cleavage site.** The target strand is labeled purple. The guide DNA is labeled green.

**Table 1 biomolecules-13-01022-t001:** A summary of the cloning methods introduced in the present review.

Cloning Method	Basic Components	Length of the Insert	Length of the Homologous Region	Cloning Efficiency	Reference
LIC-PCR	Taq poly, T4 DNA poly	150 bp–3 kb	12 bp	>90%	[8]
SLIC	Taq poly, T4 DNA poly	>3 kb	30 bp	Nearly 100% (1 insert); nearly 20% (10-fragment assembly)	[9,10]
In-Fusion	Taq poly, poxvirus DNA poly	83 bp–12 kb	15 bp or slightly longer	Up to 74%	[11]
SLiCE	bacterial extracts from RecA^−^ E. coli	Up to 19 kb	≥15 bp	Up to 99%	[15,16]
USER	uracil DNA glycosidase, T4 endo V or DNA glycosylase-lyase endo VIII		none	98%	[19,20]
Gibson assembly	T5 exo, Phusion DNA poly, Taq DNA ligase	>100 kb	Hundreds bp	>90%	[23]
TEAD	T5 exo, PEG8000	Several kb	20 bp	>90%	[30]
ET cloning	RecE/RecT or Redα/Redβ	Several kb	60 bp	>80%	[33]
iVEC	ΔhsdR ΔendAΔrecA	Several kb	20–40 bp	Upto 100%	[41]
CATCH	T5 exo, Phusion DNA poly, Taq DNA ligase	<150 kb	30 bp	20–90%	[57,58]
TAR-CRISPR	Cas9, XmaI	>100 kb	Hundreds bp	Up to 32%	[59]
ExoCET	T4 poly, Cas9, RecE/T	>100 kb	Hundreds bp	1 in 24	[60]
In vitro Packaging-Mediated Cloning	λ packaging extracts, Cas9, alkaline phosphatase	Up to 40 kb	None	Up to 54%	[61]
C-brick	Cas12a, T4 ligase	Several kb	None	25–30%	[69]
CCTL	Cas12a, Taq DNA ligase	Several kb	None	70–80%	[70]
CT5 cloning	Cas12a, T5 exo	Up to 10 kb	15–25 bp	>90%	[71]
CAPTURE	Cas12a, T4 poly, Cre-lox system	>100 kb	39 bp	Up to 100%	[73]
CAT-FISHING	Cas12a, ligase	>100 kb	>30 nt	25–70%	[74]
AREs	PfAgo, T4 ligase	Several kb	None	Up to 80%	[83]
Plasmid Maker	PfAgo, T4 ligase	Several kb	None	18–83%	[84]

## Data Availability

The data for this review article were obtained from publicly available published studies.

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
