# Peer review of "Applications of Programmable Endonucleases in Sequence- and Ligation-Independent Seamless DNA Assembly"

_biomolecules, 2023, doi:10.3390/biom13071022_

Round 1

Reviewer 1 Report

This is a very interesting review that highlights some of the most widely used molecular cloning strategies in the context of cutting-edge developments. I do not have any major issue, except for the quality of some of the schemes in the figures, as indicated below.

1. I missed the Uracil DNA Glycosylase-based clonning method (USER). While similar to LIC/SLIC, this alternative is very convenient nowadays and provides high cloning efficiency.

2. In the section 3 the authors mention transformation efficiency required form in vivo cloning. However, I guess that the given value was measured for circular, supercoiled plasmid DNA, whereas the procedures described required transformation with linear DNA that will enter the cells with lower efficiency. The provided value could be an useful standard, but it should be detailed.

3. Figure 1 is not very informative. I suggest more details schemes, indicating overlapping region required for cloning and the enzymes involved in each method.

4. Legend of Figures 2 and 4 should be improved. Please explain the nucleic acid structures and colors and include PDB Id. for the molecular structures.

5. Regarding the use of the argonaute nucleases and high temperature limitation, I'd suggest to discuss recent works that developed mesophilic DNA-guided DNA-cleaving-based cloning with CbAgo (PMIDs 32055395 and 35420131).

6. Given the current interest in this topic, this review will benefit from some practical information. I'd suggest including some references to detailed protocols that would be useful for readers to test the methods. I am referring to laboratory protocols published on protocols.io, Bio-Protocol or Methods in Molecular Biology or similar series.

7. Some sentences may need rephrasing. For instance:

Lines 150-151. This sentence sounds weird and should have a reference. Also, I missed an uppercase after the point.

Line 181. I'd suggest "However" instead of "Whereas".

Reviewer 2 Report

The authors in the article entitled: The applications of programmable endonucleases in sequence- and ligation-independent seamless DNA assembly have taken into consideration the important problem from the point of new therapies based on the CRISPR/Cas strategy. The authors in detail discussed the cons and pros of using endonucleases. The Cas and procaryotic Ago were also considered.  Even though the article is well written, readable, and valuable for a large scientific community I have a few points which should be discussed:

-        the delivery system

-        how to resolve the RNA synthetic and stability problems

-        The stability in the cytosol

-        The influence of DNA damage on endonuclease activity.

In conclusion, after the answer to the above, I recommend the article for publication in Biomolecule Journal.

Reviewer 3 Report

In their article authors reviewed current recombination-based cloning methods and described in details implementation of CRISPR/Cas9 and pAgo nucleases into these protocols.

The review is comprehensive, all mentioned approaches are described in details and discussed appropriately. I have no major concerns.

Minor points:

1)      I would appreciate to add a table summarizing all described cloning procedures, their basic components, advantages, disadvantages, etc.. This would nicely summarize the review topic and make it more understantable.

2)      Figure 3 is not cited in the text.

 I would recommend to carefully check the text for proper English grammar. There are typos present and some sentences would deserve rephrasing. Some suggestions:

-use present tense rather than past tense in following sentences: line 12, 16, 50, 53, 211, 219, 226, 237, 274, 292, 322,

-line 22: …in most cases…

-line 51: the meaning in not much clear to me

-line 61: …transformation into…

-line 63: Up to 10 DNA fragments were …

-line 81: … expressing λ prophage…

-line 95: …at the termini…

-line 97: … it makes Gibson …

-lines 140 and 141: use “that” rather than “because’

-lines 150 and 151: However, subsequent research indicated that in vivo

-lines 174 and 175: the last sentence could be deleted as its rather vague

-line 187: meaning of the word “rend” is not clear…

-line 212: … cleavage of …

-line 230: …to combine Cas9 cleavage of genomic…

-line 242: font “in vitro

-line 253: … SpRY was discovered…

-line 261: … was identified

-line 267: use “overcome” instead of “make up”

-line 274: … Cas12a are …

-line 276: In combination …

-line 282: E.coli

-line 296: … directly capture large…

-line 314: … comparison to …

-line 319: delete “sequentially”; All known pAgos contain PIWI domain.

-line 320: “N-terminal”

-line 331: … 5’-phosphorylated …

-line 334: … generate defined …

-line 341: try to rephrase word “frontend” and “backend” – the meaning is not clear

-line 352: However, it is still …

-line 358: As mentioned above …

-line 375: rephrase “indruroduced” – use rather reviewed,  summarized ?

-line 377: … study of these … 

Round 2

Reviewer 1 Report

The authors have addressed all the point I raised. I do not have further comments.